# A divide-and-conquer approach based on deep learning for long RNA secondary structure prediction: Focus on pseudoknots identification

Loïc Omnes[1,2], Eric Angel[1], Pierre Bartet[2], François Radvanyi[3], Fariza Tahi[1]*

1 Université Paris-Saclay, Univ Evry, IBISC, 91020 Evry-Courcouronnes, France, 2 ADLIN Science, 91037 Evry-Courcouronnes, France, 3 Molecular Oncology UMR144, CNRS - Institut Curie, 75005 Paris, France

* fariza.tahi@univ-evry.fr

**Data availability statement:** The source code of DivideFold, along with all the datasets used in

## Abstract

The accurate prediction of RNA secondary structure, and pseudoknots in particular, is of great importance in understanding the functions of RNAs since they give insights into their folding in three-dimensional space. However, existing approaches often face computational challenges or lack precision when dealing with long RNA sequences and/or pseudoknots. To address this, we propose a divide-and-conquer method based on deep learning, called DivideFold, for predicting the secondary structures including pseudoknots of long RNAs. Our approach is able to scale to long RNAs by recursively partitioning sequences into smaller fragments until they can be managed by an existing model able to predict RNA secondary structure including pseudoknots. We show that our approach exhibits superior performance compared to state-of-the-art methods for pseudoknot prediction and secondary structure prediction including pseudoknots for long RNAs. The source code of DivideFold, along with all the datasets used in this study, is accessible at https://evryrna.ibisc.univ-evry.fr/evryrna/dividefold/home.

## Introduction

Ribonucleic acid (RNA) is found in all living organisms and plays essential roles in cells, serving not only as a crucial intermediary that conveys genetic instructions from DNA for protein synthesis but also participating in various biological processes, such as gene expression regulation and RNA processing. The primary structure of RNA is defined by its nucleotide sequence. The secondary structure of RNA is formed from various structural motifs (stems, hairpin loops, bulge loops, internal loops, multi loops) that occur from complementary base interactions (A pairing with U, C with G, and G with U). Pseudoknots are more complex structural motifs that, unlike others, are not nested within the rest of the secondary structure. Pseudoknots are made from at least two stems that are interlaced such that the end of one stem lies between both ends of the other stem. A visual example of a pseudoknot is given in Fig 1.

this study, is publicly accessible at https://evryrna.ibisc.univ-evry.fr/evryrna/dividefold/home.

**Funding:** LO was supported by grants from Région Ile-de-France (Paris Region PhD 2022, https://www.iledefrance.fr/aides-et-appels-a-projets/paris-region-phd-2024) and Grand Equipement National de Calcul Intensif (GENCI) / Institut du Développement et des Ressources en Informatique Scientifique (IDRIS) (grant AD011014250, http://www.idris.fr/). The funders did not play any role in the study design, data collection and analysis, decision to publish, or preparation of the manuscript.

**Competing interests:** The authors have declared that no competing interests exist.

**Fig 1. Example of a Pseudoknot.** The pseudoknot is formed from two interlaced stems displayed in red and gray respectively.

Understanding RNA's secondary structure holds great importance in biology and healthcare. It unravels insights into folding dynamics, stability, and interactions with other molecules, contributing to their functional roles [1]. Enhancing our understanding of these structures can greatly improve disease diagnosis and treatment, particularly in conditions like cancer, and open avenues for novel therapeutic interventions. Messenger RNA's secondary structure, for instance, affects translation efficiency, splicing, and protein production. Finding the pseudoknots in secondary structure is especially valuable since pseudoknots shed light on the folding of RNAs in three-dimensional space, thus holding particularly useful information on its function. For example, it has been shown that a pseudoknot can lead to a deformation of the transfer RNA in the P site of the ribosome during translation and cause an interruption of the translation process [2].

Historically, RNA secondary structure prediction methods have been categorized into those based on thermodynamics, relying on either the Minimum Free Energy [3] (MFE) (mfold [4], RNAfold [5,6], RNAstructure [7], pKiss [8,9], ShapeKnots [10], LinearFold [11]), Maximum Expected Accuracy [12] (MEA) (CONTRAfold [13], CentroidFold [14], Prob-Knot [15], IPknot [16,17], BiORSEO [18]), or both (BiokoP [19]), and those based on comparative sequence analysis (RNAalifold [20], Tfold [21]). ContextFold [22] is another method that aims at refining thermodynamics parameters using structured-prediction learning. Recently, a notable evolution has occurred with the emergence of methods leveraging deep learning techniques, as shown by achievements like AlphaFold [23,24] for proteins, and several methods have been proposed for RNA secondary structure prediction. They generally either use two-dimensional (2D) Convolutional Neural Networks (CNNs) followed by a dynamic programming optimization algorithm (SPOT-RNA2 [25], 2dRNA-Fold [26], UFold [27], CNNFold [28]), or Recurrent Neural Networks (RNNs), with an optimization algorithm (RNAstateinf [29]) or without one (VLDB GRU [30]). In the first case, the CNN is used to predict a pairing probability matrix, which is then transformed into the nearest possible secondary structure thanks to the optimization algorithm. In the second case, using a RNN allows to either directly predict a secondary structure, or to predict an intermediate state which is then converted to a secondary structure through the optimization algorithm. Others use a combination of CNNs and RNNs for the pairing probability prediction (SPOT-RNA [31], MXfold2 [32]), or Transformer layers in some cases (KnotFold [33]), and then use an optimization algorithm to obtain secondary structures. RNA-par [34] is a method based on deep learning proposed recently and designed for RNA sequence partition. Its goal is not to predict the structure or the pseudoknots of the sequence, but to partition it into its different structural domains. The fragments can then be fed to an existing structure prediction model and recombined together to obtain a secondary structure for the sequence.

A study of RNA secondary structure prediction methods [35] published in 2022 shows that some of the strongest tools able to predict pseudoknots include ProbKnot [15], IPknot [16,17], pKiss [8,9], SPOT-RNA2 [25], SPOT-RNA [31] and UFold [27]. Another tool that performs well is ShapeKnots [10], but it requires SHAPE probing data in addition to the RNA sequence to reach optimal performance.

Despite these advancements, substantial challenges persist. Computational complexity is a significant obstacle, particularly in the context of long RNAs and/or pseudoknots. Indeed, dealing with long RNAs can be an issue given the exponential growth in the number of possible secondary structures. In addition, the scarcity of structure data for long RNAs is also problematic for the training of machine learning models. Furthermore, pseudoknot prediction is a challenging problem as well and it has been proven to be NP-complete [36]. As a whole, existing approaches that are able to predict pseudoknots (pKiss [8,9], ProbKnot [15], Tfold [21], ShapeKnots [10], BiokoP [19], BiORSEO [18], SPOT-RNA [31], SPOT-RNA2 [25], UFold [27], KnotFold [33]) require a time complexity in at least $O(n^2)$ or higher, which makes it difficult to handle long RNAs. One exception to this is IPknot [16,17], which recently introduced a new version [17] with a linear time complexity in $O(n)$.

We introduce a sequence partition strategy with a linear time complexity in $O(n)$ based on deep learning techniques in order to enhance the accuracy and efficiency of long RNA structure prediction including pseudoknots. The core concept driving our approach is to harness the strengths of high-performing models optimized for small RNAs while addressing their limitations in handling extended sequences. Through a recursive partition, long RNAs are separated into smaller fragments, facilitating the application of established prediction models designed for shorter RNAs. We present here DivideFold, an extension of our previous work [37]. We have previously shown encouraging preliminary results. Since then, we have extended our work for the prediction of pseudoknots. We have also investigated several improvements, including (i) enhancing the quality of our model through a new architecture and optimized hyperparameters, (ii) significantly reducing the computation time of our model by eliminating less significant input features, (iii) reviewing our dataset to avoid structural biases and achieve more meaningful data splits, (iv) exploring metrics of interest for quantifying the quality of sequence partitions and (v) implementing a new data augmentation step based on nucleotide mutation.

The paper is organized as follows. We first present our approach and describe the details of our divide model's implementation and its training procedure. We then give a summary of the existing works related to ours. Next, we detail the datasets and methods used in our experiments and we report our results for pseudoknot prediction and secondary structure prediction including pseudoknots. Finally, we discuss the results and opportunities for future improvements.

## Method

### Partition strategy

Our goal in the proposed approach is to partition a long sequence into shorter, structurally independent fragments to make use of existing structure prediction models designed for short RNAs and predict their structures. Then, the predicted structures for the different fragments are recombined together to their original locations in the sequence and constitute the global structure prediction for the long sequence. Our approach consists of any combination of a divide model and a structure prediction model. The workflow of our method, called Divide-Fold, is depicted in Fig 2. While we design the divide model ourselves, we choose the structure model among already existing secondary structure prediction models. Our approach achieves a time complexity in $O(n)$ in regards to the input sequence length, by using a one-dimensional (1D) CNN divide model for the partition step, and by ensuring that every fragment is shorter than a chosen length for the structure prediction step. Firstly, the 1D CNN divide model ensures that the partition step is in $O(n)$ since the kernel size is constant across layers, and uses decreasing dilation rates to capture long-range relationships in the sequence

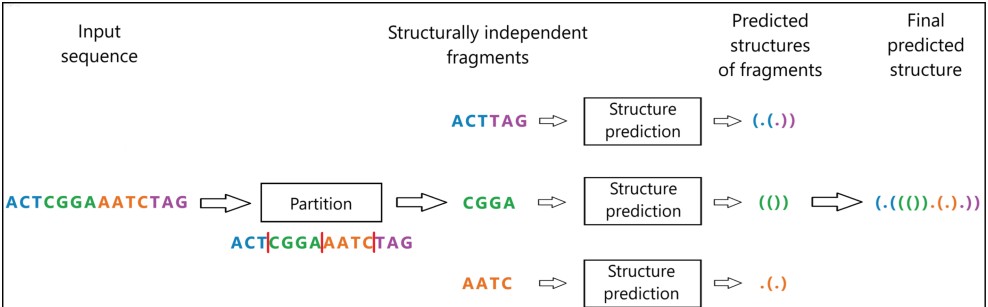

**Fig 2. DivideFold workflow.** The sequence is partitioned into several fragments and the different predicted secondary structures are then recombined to the original positions of the fragments in the sequence.

while avoiding the need to evaluate all possible pairs. Secondly, having a constant maximum length for the fragments guarantees that the structure prediction step is also in $O(n)$, regardless of the structure prediction model used. Only the feature extraction step is actually not in $O(n)$ because of the motifs insertion using regular expressions, but this step is very fast in comparison to the other steps in the algorithm, meaning that our approach reaches a linear time complexity in $O(n)$ in practice.

A key challenge in partitioning the sequence lies in strategically selecting cut points as to preserve the structural integrity of the RNA as much as possible. Indeed, any base pair occurring between two different fragments will be broken and impossible to recover at the structure prediction step, since each fragment is processed separately. To avoid this, cut points should ideally be placed in between the different stems that are minimally nested (excluding pseudoknots). This would result in fragments where all base pairs only occur within themselves, meaning that no base pairs are broken during the process except pseudoknots.

Once the cut points have been chosen, we allow the immediate combination of the leftmost and right-most parts into a single fragment, creating an artificial fragment that, while not directly continuous in the original sequence, maintains structural coherence. This step is crucial to make it possible to partition further a sequence where a long-range stem contains a large portion of the structure without breaking that stem. Indeed, in this scenario, the stem would inevitably be broken if only continuous fragments are used in the partition. Instead, by combining the external parts in a discontinuous fragment, the sequence is guaranteed to be partitioned into at least two fragments, ensuring that it will be shortened, while having the opportunity to preserve structural information. We decide to repeat choosing cut points and merging outer parts recursively to allow to focus on the minimally nested regions at each iteration and progressively partition the sequence until all fragments are shorter than a chosen length. This is important because some regions may be more deeply nested than others. The precision of cut point selection is an important issue, as any misstep in this process can significantly compromise the accuracy of the final predicted structure over the iterations. Fig 3 provides a visual representation of one partition iteration.

## Finding pseudoknots

Formally, a pseudoknot is made of at least two nested groups of base pairs where each base pair $(i, j)$ in the first group and each base pair $(k, l)$ in the second group are such that $i<k<j<l$. An example is shown in Fig 1.

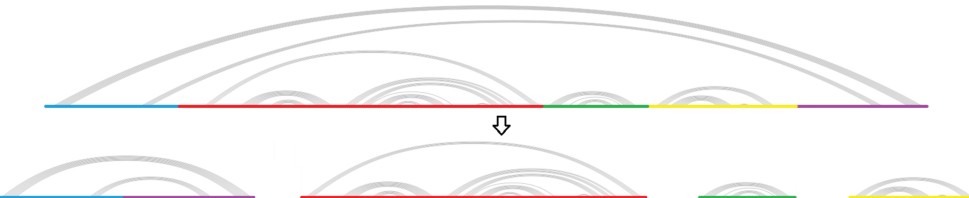

**Fig 3. One partition iteration.** The partition process occurs recursively until all fragments are shorter than a chosen length. At each iteration, the left-most and right-most parts are combined to form a single fragment. Note that the red fragment may be further partitioned similarly at the next iteration.

Pseudoknots are removed for generating ideal cut points. Notice however that by using a structure prediction model that is able to predict pseudoknots, it is possible to find pseudoknots at the structure prediction step, as long as they are located within a single fragment (whether it is continuous in the original sequence or an artificial fragment made of several disconnected parts). This is depicted in Fig 4. In this example, the RNA sequence includes five pseudoknots. Four of them can be recovered because they occur the same fragment.

## Divide model

In this section, we present our algorithm for predicting cut points in a given sequence, using only the sequence information. Features are first extracted from the sequence and given to the divide model. The model then generates cutting probabilities for each nucleotide in the sequence. Cut points are finally selected from the predicted probabilities through a peak detection algorithm. This process is repeated recursively until the fragments are shorter than a chosen length.

**Feature extraction.** For the feature extraction, we chose a concatenation of one-hot embeddings and insertion of RNA structure motifs. One-hot embedding is the most standard and direct way to encode the sequence of nucleotides, without additional information. RNA structure motifs insertion involves integrating known motifs or structural patterns into the RNA sequence as additional features. To do this, we use the motifs from the CaRNAval [38] database. For each motif, we check for matches in the sequence through regular expressions and count the number of possible configurations for the motif to be inserted at each point in the sequence. This is illustrated in Fig 5. It should be noted that some of the motifs require a considerable amount of time to be inserted and bring little extra information because they

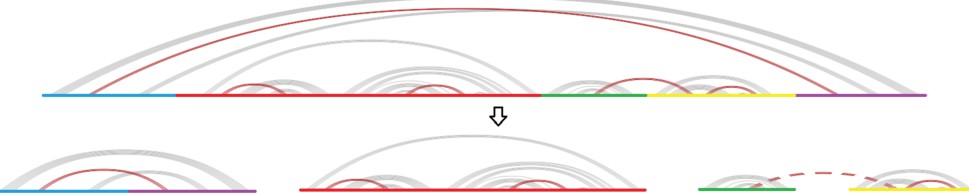

**Fig 4. Pseudoknot conservation during partition.** This is an example of a partition iteration where the RNA sequence includes five pseudoknots, displayed in red. Four of them can be recovered because they occur in the same fragment, continuous or not. However, the pseudoknot represented in a dashed line occurs between two different fragments (in green and yellow) and will be lost.

Insertions of the
motif UC*GA

```
GUUCGGAUUGAGGGUCUGCAACUCGACC
GUUCGGAUUGAGGGUCUGCAACUCGACC
GUUCGGAUUGAGGGUCUGCAACUCGACC
GUUCGGAUUGAGGGUCUGCAACUCGACC
GUUCGGAUUGAGGGUCUGCAACUCGACC
```
⇩ sum ⇩

Motif feature   0 0 3 3 0 1 1 0 0 1 1 0 0 1 1 0 0 0 0 0 0 0 1 1 3 3 0 0

**Fig 5. Illustration of the motif insertion.** The insertion of the motif UC*GA in a sequence is shown here. The possible configurations of the motif are found inside the sequence using regular expressions. Then, at each nucleotide in the sequence, the number of configurations that occur at that position are summed, leading to a feature vector. This process is repeated for each motif to be inserted, each time yielding a different feature for the sequence.

are highly correlated to other lighter motifs. For this reason, we choose to remove the heavier motifs and observe a drastic reduction of the computation time at no expense for the performance.

**Deep neural network architecture.** Our divide model computes a cutting probability for each nucleotide in the sequence, using a deep neural network. The architecture of the network is shown in Fig 6. We use a 1D dilated CNN encoder made of 10 layers with kernel size 3, dilation rates decreasing in powers of 2 and ReLu activations. This convolutional encoder is then followed by a fully connected regressor with a sigmoid activation to compute cutting probabilities at every nucleotide position. With this architecture, the divide model has a time complexity in $O(n)$.

**Peak detection.** Following the prediction of cutting probabilities at each point in the sequence by the model, the signal.find_peaks algorithm from the Scipy [39] library is employed to detect peaks in the probabilities and select the cut points that will be chosen to partition the sequence. Depending on the probabilities predicted by the model, the peak detection algorithm may retain an arbitrary number of cut points. We ensure that at least two cut points are selected, so that the sequence is guaranteed to be partitioned into at least two fragments.

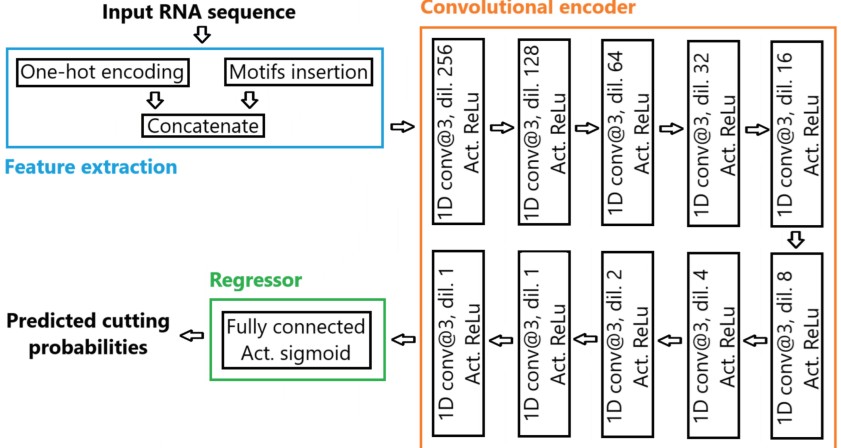

**Fig 6. Architecture of the divide model.** A 1D dilated CNN is used, followed by a fully connected layer with a sigmoid activation. Convolutional layers use a kernel size of 3, dilation rates decreasing in powers of 2 and ReLu activations.

Fig 7 illustrates the predicted cutting probabilities and the cut points chosen by the peak detection algorithm for the *Bradyrhizobium* 16S ribosomal RNA gene. In this example, the sequence is partitioned into five parts, with the left-most and right-most parts being combined into a single fragment, resulting in four fragments. We also show the structure of the RNA for reference. It can be seen that the model correctly cuts the sequence in between the different parts, and no base pairs are broken. The base pairs connecting the left-most and right-most parts are also conserved since these two parts are combined.

**Training labels and data augmentation.** We generate training labels for cut points at ideal positions, from the RNA's sequence and its secondary structure. Specifically, we remove pseudoknots from the structure and we position the cut points in between the different minimally nested stems. This is only possible once the pseudoknots have been removed, which is the reason for this. These cut points are guaranteed to preserve all the base pairs in the structure, making them relevant training labels for our divide model. Because the partition process is recursive, at each iteration the current sequence to be partitioned is used as input and new training labels are generated. This means that for each sequence in the training dataset, each iteration when partitioning to generate training labels contributes to a new observation for the divide model's training, thus serving as a valuable data augmentation tool. This is particularly advantageous for long sequences, where the number of iterations is substantial. We also apply a data augmentation procedure to every input sequence in our training, where mutations are randomly applied to between 0% and 10% of the base pairs in the sequence. When a base pair is mutated, we modify it to another random canonical base pair (AU, GC, or GU) in random order. The training of the model employs the Adam optimizer [40]. To remove pseudoknots, we use functions that have been added to the PyCogent [41] project. There are several different ways of removing pseudoknots, depending on the criterion used [42]. After training and evaluating our model using the different possible criteria for pseudoknot removal, we choose to use the IR [42] method (incremental, range) that prioritizes keeping long-range stems.

**Loss function.** We adopt a distinctive loss function for the divide model, denoted as *L*. We design a customized ground truth function *y* (Eq (1)) and we set *L* as the squared error

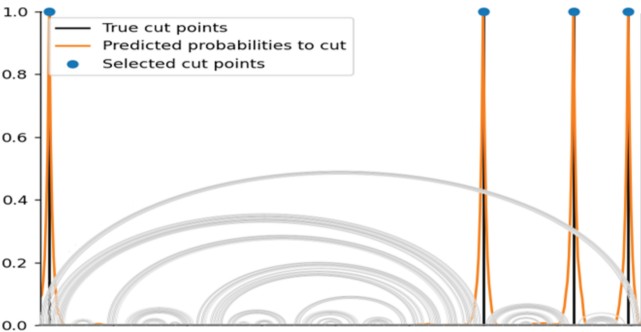

**Fig 7. Cut points prediction.** The *Bradyrhizobium* 16S ribosomal RNA is given here as an example. The divide model predicts cutting probabilities, and cut points are then selected using a peak detection algorithm. The selected cut points can be seen in the blue dots, and ideal positions in the black bars. The structure of the RNA is displayed for reference. No base pairs are broken here.

between $y$ and the predicted probabilities $\hat{y}$ (Eq (2)):

$$\forall i \in [\![1\,;\,n]\!], y_i(C) = \exp^{-\lambda * \min_{c \in C} |i - c|} \tag{1}$$

$$L(\hat{y}, C) = \sum_{i=1}^{n} (\hat{y}_i - y_i(C))^2 \tag{2}$$

where: $\hat{y}$ is the predicted cutting probabilities at each point in the sequence, $C$ is the set of target cut points, $n$ is the sequence length and $\lambda > 0$ is a hyperparameter relative to how steeply the target score function decreases when getting farther from the labeled cut points.

The training process focuses on optimizing the model parameters to minimize the loss function, which quantifies the differences between the positions of predicted and target cut points. Rather than assigning a value of 1 for an exact prediction of the position of a target cut point and 0 otherwise, the custom function we designed uses an inverse exponential distance to the nearest target cut point. This provides a much more extensive support, with non-zero values everywhere, enabling a more reliable flow of information back to the model. This function emphasizes precise cut point positioning, which is crucial since even minor deviations can accumulate over the iterations, significantly influencing the final predicted RNA structure. Overall, this loss function facilitates the training of the model, offering less sparse information and demonstrating superior performance compared to standard loss functions. Like other hyperparameters, $\lambda$ was experimentally determined through a grid search on our Validation dataset. We chose a value of 0.5 for $\lambda$.

## Related work

We consider here two challenges for the prediction of RNA secondary structure: predicting pseudoknots, and handling long sequences.

Numerous methods have been proposed in the literature for predicting RNA secondary structure including pseudoknots. Several are based on thermodynamics. pKiss [8,9] is an algorithm that focuses on finding kissing hairpin pseudoknots by minimizing MFE through dynamic programming, but its computation time is rapidly growing in regards to the sequence length. ShapeKnots [10] uses a dynamic programming algorithm to minimize MFE and predict secondary structures including pseudoknots. On the downside, Shape-Knots requires SHAPE data to reach its full potential, and can only be used for short RNAs. BiokoP [19] is another method capable of predicting pseudoknots by optimizing both MFE and MEA and has shown better results than pKiss and IPknot. However, BiokoP can only be used for short RNAs. BiORSEO [18] uses a combination of MEA optimization and motifs detection to find pseudoknots, but it can only be used for short RNAs due to high computation time and memory usage.

As for deep learning methods, SPOT-RNA [31] is an ensemble of models based on a combination of CNN and RNN layers to predict a secondary structure including pseudoknots. SPOT-RNA is intended for RNAs shorter than 500 nt. SPOT-RNA2 [25] is an improvement of SPOT-RNA, based on an ensemble of dilated CNN models. SPOT-RNA2 is able to predict secondary structures including pseudoknots for RNAs no longer than 1,000 nt. KnotFold [33] is a recent model that is able to predict pseudoknots. It uses a minimum-cost flow algorithm on a potential function that is learned through an attention-based neural network. KnotFold has shown strong results for both pseudoknotted and non-pseudoknotted base pairs.

Considering the prediction of RNA secondary structure for long sequences, Linear-Fold [11] is a low complexity model based on an enhancement of RNAfold [5,6] to improve

the computation time. It has a linear time complexity in $O(n)$, allowing it to process very long RNAs. However, LinearFold is not capable of predicting pseudoknots.

Some approaches have attempted to both predict pseudoknots and handle longer RNA sequences. ProbKnot [15] aims to maximize MEA instead. It is capable of processing long sequences, hence at a considerable time cost in a $O(n^2)$ time complexity. Tfold [21] is a tool able to search for stems and pseudoknots recursively in long RNAs by optimizing criteria of stability, conservation and covariation in sequences that are selected from a large set of homologous sequences. Tfold is based on a divide-and-conquer algorithm with a time complexity in $O(n^2)$. However, it is needed to know homologous sequences in order to use it, which is rare in the case of long RNAs. IPknot [16,17] is based on integer programming to maximize MEA and predict RNA secondary structures including pseudoknots. A new version [17] released in 2022 and using base pairing probabilities from LinearPartition [43] allows to deal with long RNAs with a time complexity in $O(n)$ instead of $O(n^3)$. UFold [27] is a CNN-based deep learning method that uses a U-Net model to predict a contact map of pairing probabilities, which is then converted to a secondary structure, including pseudoknots, through a postprocessing algorithm.

Finally, RNA-par [34] is different from structure prediction tools in that it predicts a partition instead to subdivide an RNA sequence into shorter fragments, similarly as in our approach. It is then possible to obtain a predicted structure for the sequence by using an existing structure prediction model on the fragments and recombining them together. However there are some key differences with our work. (i) RNA-par does not allow external parts to be merged, yielding only continuous fragments. Thus, any long-range stem will cause a large region to be impossible to partition without losing that stem. This will likely happen often, given that these long-range stems are common motifs in RNA structure. (ii) The partition process of RNA-par is only made of one iteration. Yet, more nested structures will require to repeat steps in order to partition deeper while preserving the structure. Thus, it is impossible to further partition large nested structures in a single iteration without breaking base pairs. Because of this, large regions in the sequence may be left to the structure prediction model, instead of short fragments as intended. (iii) RNA-par does not remove pseudoknots beforehand when building training labels, which can also cause large regions to be linked and impossible to partition without breaking base pairs. This can lead to the absence of positive labels for cut points in large regions and the model's inability to process such regions as a whole. (iv) RNA-par is solely trained and tested on RNAs shorter than 200 nucleotides. In contrast, we aim to focus on much longer RNAs in our approach.

## Results

### Dataset

We used the bpRNA-1m [44] database, a dataset featuring over 100,000 RNA sequences alongside their corresponding secondary structures, many of which are longer than 1,000 nt. On the downside, bpRNA-1m contains very few RNAs longer than 1,600 nt, which could make the training of our model difficult for long RNAs past this threshold, but this seems to be the case of every RNA secondary structure dataset. The structures in the bpRNA-1m database are predominantly determined through comparative sequence analysis from the Comparative RNA Web (CRW) database [45]. The ArchiveII [46] and RNAStralign [47] datasets are commonly used for RNA secondary structure, but we did not consider them since they contain almost only short RNA sequences and the long RNAs that they contain already have their families included in bpRNA-1m. We clustered the bpRNA-1m dataset using CD-HIT-EST [48] at a 80% sequence similarity threshold. We then split it into a Train, Validation

and Test datasets according to the clusters, ensuring a similarity no higher than 80% between the sequences in the three datasets. We use the Train dataset for the training of our model. The Validation dataset is used for the choice of the maximum fragment length. Lastly, we use the Test dataset for evaluating the performances of our method and comparing it to the state-of-the-art in our experiments. We only keep the pseudoknotted structures in the Test dataset for our experiments. Our final Train, Validation and Test datasets contain 57,251, 2,544 and 4,334 RNAs respectively.

## Benchmarked methods

We conduct a benchmark comparing DivideFold to some of the different most effective tools from the literature that are capable of predicting pseudoknots. We consider in our experiments five methods from the literature: IPknot [16,17] (version 1.1.0), ProbKnot [15] (version 6.4), KnotFold [33], pKiss [8,9] (version 2.2.12) and UFold [27]. We do not add Shape-Knots [10], BiokoP [19], BiORSEO [18], SPOT-RNA [31] and SPOT-RNA2 [25] to our benchmark since they can only handle short RNAs. We also disregard Tfold [21] since we would need to know homologous sequences, and LinearFold [11] since it does not predict pseudoknots. Finally, we could not include RNA-par [34] in our benchmark because we were unable to use the code provided in the website mentioned in the article.

All methods (IPknot, ProbKnot, KnotFold, pKiss and UFold) are used with their default parameters. We re-train UFold on the Train dataset. In the case of KnotFold, we use the weights pre-trained on the TR0 [32] dataset. When designing our dataset, we make sure that the Train dataset includes the TR0 dataset, sparing us the need to re-train KnotFold.

## Evaluation of structure prediction models

We first evaluate the different benchmarked methods to decide which one we will integrate in our approach as the structure prediction model. To that extent, we evaluate IPknot, ProbKnot, KnotFold, pKiss and UFold on the Test dataset for short sequences below 1,000 nt, since we will use the structure prediction model on short fragments. We measure the models' performance for the secondary structure prediction including pseudoknots through the precision, recall and F-score metrics:

$$Precision = \frac{TP}{TP + FP} \qquad Recall = \frac{TP}{TP + FN} \tag{3}$$

$$F\text{–}score = \frac{2 * Precision * Recall}{Precision + Recall} \tag{4}$$

where *TP*, *FP*, *FN* are the number of true positive, false positive and false negative base pairs respectively. Formally, for a sequence of length $n$, among the $\frac{n*(n-1)}{2}$ possible base pairs, when considering the secondary structure, a base pair is positive if a bond is predicted between the two bases, and negative if no bond is predicted. The base pair is considered either true or false depending on whether the prediction is actually right or not.

Fig 8 shows the prediction performances as well as the computation times for each of the five tools. UFold only accepts sequences shorter than 600 nt. KnotFold has the highest performance out of the five methods. It reaches a mean F-score for secondary structure prediction including pseudoknots that is substantially better than that of the other four methods on our dataset, for sequences in the 400-599 nt, 600-799 nt, and 800-999 nt ranges. It is also among

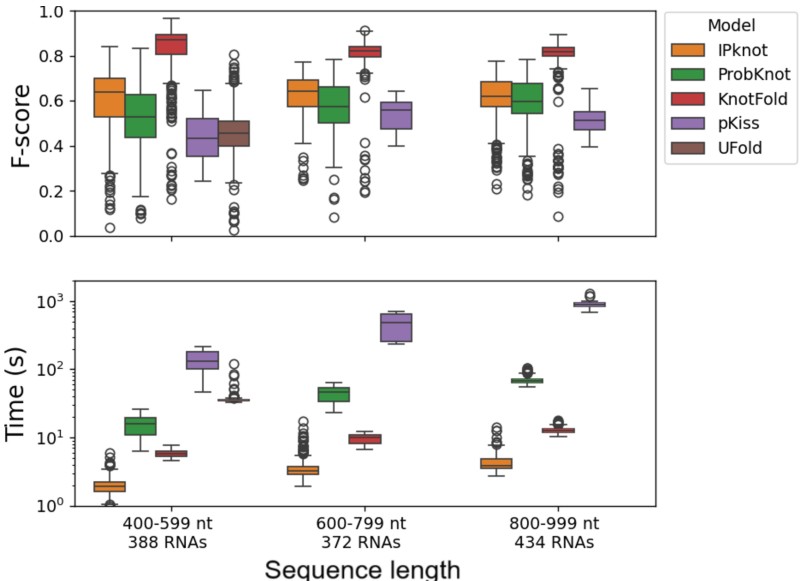

**Fig 8. Structure prediction models evaluation by sequence length.** The F-score for secondary structure prediction including pseudoknots depending on the input sequence length for RNAs shorter than 1,000 nt is shown for IPknot [16,17], ProbKnot [15], KnotFold [33], pKiss [8,9] and UFold [27] on the Test dataset. The computation times are displayed for reference in a log scale.

the fastest tools in our benchmark along with IPknot. In view of these results, we choose KnotFold as the structure prediction model in our approach.

## Maximum fragment length hyperparameter

The maximum length of a fragment is a crucial hyperparameter in our method. If the maximum fragment length is too low, the fragments will be shorter, but at the cost of a higher risk of breaking base pairs, ultimately leading to a drop in performance overall. On the other hand, if the maximum fragment length increases, less base pairs will be broken during partition, but the resulting fragments will be longer, making the structure prediction task harder.

In order to determine the optimal value for the maximum fragment length, we introduce two metrics of interest for measuring the quality of the divide model:

- the break rate, which is the percentage of base pairs that are broken during partition, excluding pseudoknots.
- the compression rate, which reflects the factor by which the fragments are shortened compared to the original sequence:

$$compression = 1 - \sum_{i=1}^{p} \left( \frac{n_i}{n} \right)^2 \tag{5}$$

where $n_1, ..., n_p$ are the lengths of the final fragments and $n$ is the sequence length, such that $\Sigma_{i=1}^{p} n_i = n$

Generally, predicting the structure of a shorter sequence is easier, meaning that the compression rate should be as high as possible. At the same time, the break rate should be as low

as possible, so as to preserve the structure. The two metrics are closely linked because a higher compression rate is due to a deeper partition and will generally lead to a higher break rate. We evaluate our approach for different values of the maximum fragment length up to 1,200 nt on the Test dataset. We cannot consider higher values for the maximum fragment length past 1,200 nt because most RNAs in the Test dataset are either shorter than this threshold or only slightly longer. Since sequences shorter than the maximum fragment length cannot be partitioned, raising the threshold further would prevent us from partitioning nearly all of these sequences. To ensure that we compare the results on the same data for the different values of the maximum fragment length, we only consider here sequences longer than 1,200 nt. This is because shorter RNAs would not be partitioned for the higher values of the maximum fragment length, and all values would not be evaluated on the same data.

We display in Fig 9 the mean compression rate and break rate of our divide model for different values of the maximum fragment length. The F-score for secondary structure prediction is also indicated for reference. The F-score is highest when the maximum fragment length is between 600 nt and 1,000 nt, even though the variations in F-score are slim. At 1,000 nt for the maximum fragment length compared to 800 nt, the compression rate decreases (71.3% instead of 72.4%) but the break rate is also lower (2.2% instead of 2.5%).

To choose a value for the maximum fragment length, we perform a more detailed analysis. We measure the F-score for the secondary structure prediction including pseudoknots and we separate the sequences from the Test dataset into groups according to their lengths. We show the results in Fig 10. The performance is slightly better for a maximum fragment length of 800 nt compared to 1,000 nt for RNAs shorter than 1,400 nt, but it is slightly worse for RNAs longer than 2,000 nt. As a whole, it appears that a maximum fragment length of 1,000 nt yields the best performance overall when considering the various RNA sequence length ranges studied.

In light of these results, we choose a value of 1,000 nucleotides for the maximum length of a fragment in our approach. Therefore, RNAs shorter than that threshold will not be partitioned in our approach. Because of this, we focus on RNAs longer than 1,000 nt in the rest of our experiments.

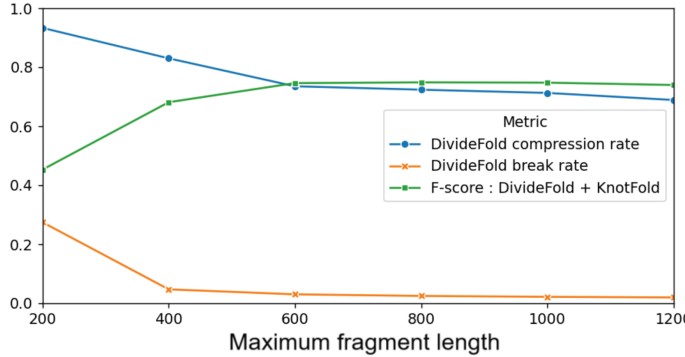

**Fig 9. Compression and break rates with respect to maximum fragment length.** Mean compression rate and break rate of our divide model for different values of the maximum fragment length hyperparameter, for sequences longer than 1,200 nucleotides on the Test dataset. The F-score for secondary structure prediction is shown for reference.

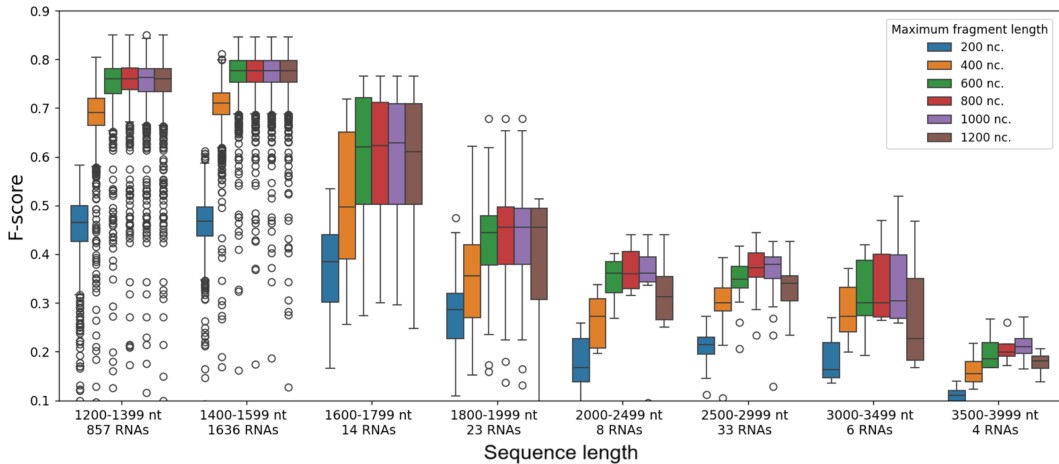

**Fig 10. Secondary structure F-score for DivideFold with respect to maximum fragment length, by sequence length.** The F-score for secondary structure prediction including pseudoknots depending on the input sequence length for RNAs longer than 1,200 nt is shown for DivideFold, using KnotFold as the structure prediction model, for different values for the maximum fragment length, on the Test dataset.

## Pseudoknot prediction results

To assess our capability to predict pseudoknots, we compare DivideFold to IPknot [16,17], ProbKnot [15] and KnotFold [33] on the Test dataset for sequences longer than 1,000 nt. UFold [27] could not be evaluated on the Test dataset because it only accepts RNAs shorter than 600 nt, and pKiss [8,9] could not be evaluated either on the Test dataset because its time complexity prevented us from testing it on RNAs longer than 1,000 nt in our experiments. KnotFold could not process RNAs longer than 2,500 nt in our experiments because of memory overflows.

To find the pseudoknots in the structure, we look for any two nested groups of base pairs where base pair $(i, j)$ in the first group and base pair $(k, l)$ in the second group are such that $i<k<j<l$. Furthermore, we consider that two pseudoknots are similar if they share at least one common base pair in their first groups and one common base pair in their second groups. A predicted pseudoknot is considered a true positive if there is a similar pseudoknot in the real structure, and a false positive otherwise. Real pseudoknots with no similar pseudoknots in the predicted structure are false negatives. We use the precision, recall and F-score metrics when evaluating pseudoknot prediction.

We show the mean pseudoknot prediction results in Table 1. IPknot and ProbKnot show poor results when predicting pseudoknots. KnotFold performs much better and manages to find 45.6% of the pseudoknots on average, although it also predicts many false positives, hence a low precision. In contrast, our approach allows to find up to 81.4% of the pseudoknots on average and increases precision by a factor of 21 compared to KnotFold.

We also inspect the pseudoknot prediction performance depending on the RNA sequence length. We separate the sequences from the Test dataset into groups according to their lengths and we show the results in Fig 11. Our approach exhibits superior performance compared to the benchmarked methods. However, DivideFold still struggles to accurately predict the pseudoknots of long RNAs beyond 2,500 nt. This could be due to the very limited amount of available training data for RNAs longer than 1,600 nucleotides, which could have hampered our ability to derive reliable results in our experiments.

**Table 1. Pseudoknot prediction performance.**

| Model | Recall | Precision | F-score |
|---|---|---|---|
| DivideFold + KnotFold | **0.814** | **0.064** | **0.117** |
| KnotFold | 0.456 | 0.003 | 0.006 |
| IPknot | 0.008 | 0.001 | 0.002 |
| ProbKnot | 0.002 | 0.001 | 0.001 |

The performance of DivideFold, IPknot [16,17], ProbKnot [15] and KnotFold [33] is reported here for RNAs longer than 1,000 nt on the Test dataset.

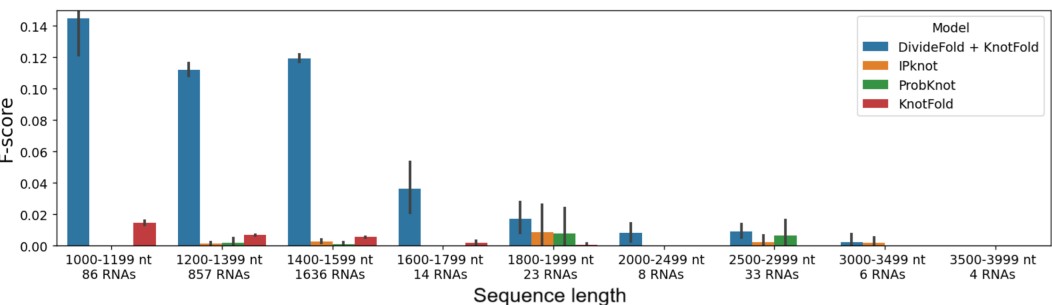

**Fig 11. Pseudoknot prediction performance by sequence length.** The F-score is shown for pseudoknot prediction depending on the input sequence length for RNAs longer than 1,000 nt. Confidence intervals are shown in the black bars. The performance of DivideFold, IPknot [16,17], ProbKnot [15] and KnotFold [33] is reported here on the Test dataset. Missing bars mean that the performance is very close to zero and cannot be seen.

## Secondary structure prediction results including pseudoknots

We evaluate here the ability of DivideFold to predict the secondary structure of long RNAs including pseudoknots. For this analysis, we also compare DivideFold to IPknot [16,17], ProbKnot [15] and KnotFold [33] on the Test dataset for sequences longer than 1,000 nt. UFold [27] and pKiss [8,9] could not be evaluated on the Test dataset, and KnotFold could only process RNAs longer than 2,500 nt, for the same reasons stated before. We show the results in Table 2. Our approach demonstrates better recall, precision and F-score on average compared to IPknot, ProbKnot and KnotFold.

We also display the secondary structure prediction performance including pseudoknots depending on the RNA sequence length in Fig 12. Our approach exhibits better performance for RNAs up to 2,500 nucleotides long compared to the other methods. Yet, as for the pseudoknots, DivideFold fails to predict precisely the secondary structures of longer RNAs. This is also probably caused by the lack of training data for long RNAs past 1,600 nt. IPknot shows better results for sequences longer than 2,500 nt.

**Table 2. Secondary structure prediction performance including pseudoknots.**

| Model | Recall | Precision | F-score |
|---|---|---|---|
| DivideFold + KnotFold | **0.737** | **0.747** | **0.740** |
| IPknot | 0.526 | 0.564 | 0.542 |
| ProbKnot | 0.470 | 0.431 | 0.448 |
| KnotFold | 0.512 | 0.324 | 0.397 |

The performance of DivideFold, IPknot [16,17], ProbKnot [15] and KnotFold [33] is reported here for RNAs longer than 1,000 nt on the Test dataset.

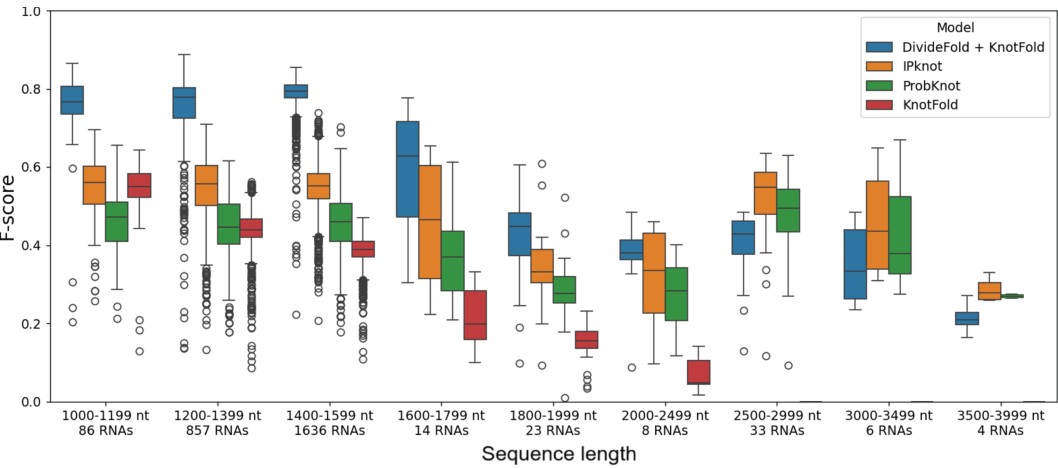

**Fig 12. Secondary structure prediction performance including pseudoknots by sequence length.** The F-score is shown depending on the input sequence length for RNAs longer than 1,000 nt. The performance of DivideFold, IPknot [16,17], ProbKnot [15] and KnotFold [33] is reported here on the Test dataset.

## Time and memory cost

We run our experiments on the Jean Zay supercomputer using one V100 GPU with 16 Go of memory and we measure the computation time. Our approach remains fast and ensures scalability to long RNAs, as evidenced in Fig 13, addressing the computational issues faced by many models. By partitioning the sequences into shorter fragments, our approach allows to decrease significantly the computation time. This is because pseudoknot prediction algorithms often have a time complexity in $O(n^2)$ or higher, meaning that it is faster to process several shorter sequences rather than one large sequence. Furthermore, the partition step is very fast (less than 0.3 seconds on average on the Test dataset). Thus, the computation time

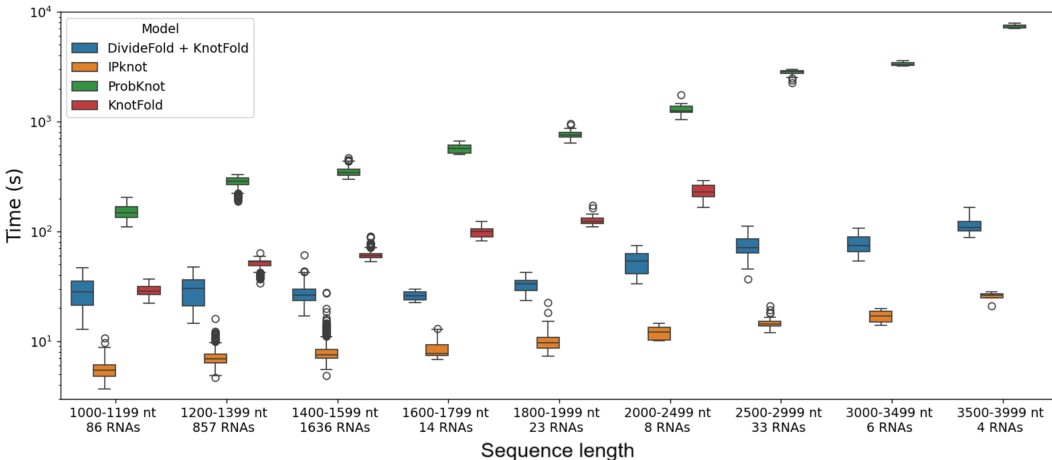

**Fig 13. Computation time by sequence length.** The time is displayed in a log scale depending on the input sequence length for DivideFold, IPknot [16,17], ProbKnot [15] and KnotFold [33] for RNAs longer than 1,000 nt on the Test dataset.

is almost only due to the structure prediction model, and not the divide model. Notably, IPknot [16,17] is very fast in its new version [17].

Memory overflows can also be problematic for long RNAs for other tools, especially in the case of CNN-based deep learning methods. Indeed, these approaches involve convolutional layers and large matrices with a space complexity growing in $O(n^2)$ which eventually becomes intractable in regards to the memory. However, our approach allows to avoid memory overflows by dealing only with short fragments.

## Case study: the 16S ribosomal RNA

The 16S ribosomal RNA is of significant importance and was extensively studied throughout the years. It has a length of 1,512 nucleotides and its structure is well-known. To showcase a practical example, we display its structure (which includes pseudoknots) in Fig 14 and we highlight the final cut points selected by our divide model. The cut points are shown in red dots. It can be seen that the RNA is accurately partitioned into its different domains. Our divide model does not break any base pairs excluding pseudoknots. Moreover, it requires only 0.2 seconds to compute.

The pseudoknot prediction results for this RNA are shown in Table 3. In this experiment we were able to use pKiss [8,9], even though it takes a long time to compute, since there is only one sequence to process here. DivideFold manages to find two pseudoknots out of three, while KnotFold finds one, and IPknot, pKiss and ProbKnot find none.

We can see in Fig 14 that two out of the three pseudoknots (the one on the left and the one at the top right) in the structure are located within the same fragment and can be recovered after the partition. As a matter of fact, both are then successfully predicted at the structure prediction step. However, the third pseudoknot (at the bottom right) occurs between two different fragments (the pseudoknot takes place between the left-most part and an internal part, which are not combined together and yield to two separate fragments) and is irrecoverable.

The results for the secondary structure prediction of the 16S ribosomal RNA are shown in Table 4. DivideFold yields a good performance, surpassing that of IPknot, ProbKnot, KnotFold and pKiss in terms or recall, precision and F-score, while remaining fast and eliminating the risk of memory overflows. Even though our approach takes 19 more seconds to compute

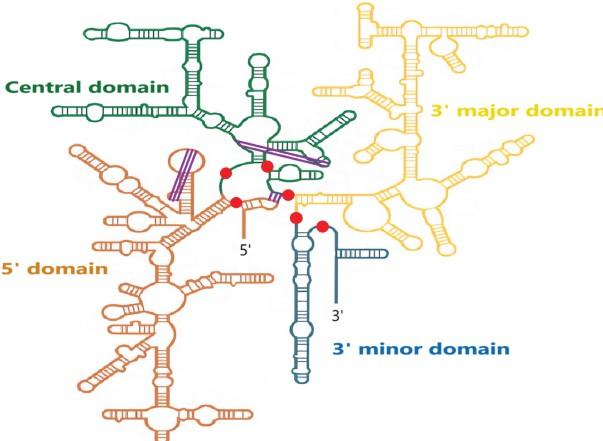

**Fig 14. Secondary structure of the 16S ribosomal RNA [49].** The final cut points chosen by our divide model are shown in the red dots. The pseudoknots are displayed in purple.

**Table 3. Results for pseudoknot prediction on the 16S ribosomal RNA.**

| Model | Pseudoknots found | False positives |
|---|---|---|
| DivideFold + KnotFold | **2 / 3** | 14 |
| KnotFold | 1 / 3 | 367 |
| ProbKnot | 0 / 3 | **4** |
| IPknot | 0 / 3 | 11 |
| pKiss | 0 / 3 | 17 |

The performance of DivideFold, IPknot [16,17], ProbKnot [15], KnotFold [33] and pKiss [8,9] is reported here. It is shown how many of the three pseudoknots are correctly found and how many false positive pseudoknots are incorrectly predicted.

**Table 4. Results for secondary structure prediction including pseudoknots on the 16S ribosomal RNA.**

| Model | Recall | Precision | F-score | Time |
|---|---|---|---|---|
| DivideFold + KnotFold | **0.770** | **0.848** | **0.807** | 0:00:27 |
| IPknot | 0.443 | 0.528 | 0.482 | **0:00:08** |
| ProbKnot | 0.376 | 0.392 | 0.384 | 0:06:33 |
| KnotFold | 0.441 | 0.315 | 0.368 | 0:01:10 |
| pKiss | 0.277 | 0.269 | 0.273 | 2:37:09 |

The performance of DivideFold, IPknot [16,17], ProbKnot [15], KnotFold [33] and pKiss [8,9] is reported here, as well as the computation time.

compared to IPknot, the prediction is significantly better. Compared to the other methods, pKiss has the lowest performance and also takes a long time to compute.

## Study of RNAs shorter than 1,000 nt

We assessed in the previous sections the performance of DivideFold for RNAs longer than 1,000 nt on the Test dataset. Here, we evaluate DivideFold's abilities for RNAs shorter than 1,000 nt on the Test dataset. We compare DivideFold to IPknot, ProbKnot, KnotFold, pKiss and UFold on the Test dataset for sequences between 500 nt and 1,000 nt. A maximum fragment length of 500 nt was used for DivideFold in this experiment.

We show the results depending on the RNA sequence length in Fig 15 and Fig 16 for the pseudoknot prediction, and the structure prediction including pseudoknots respectively.

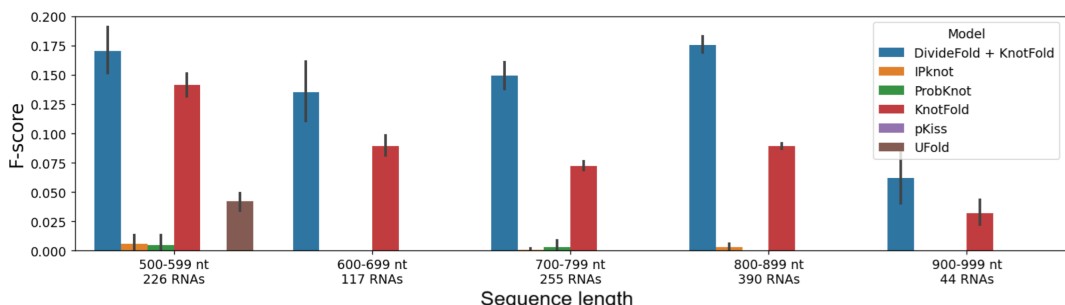

**Fig 15. Pseudoknot prediction performance by sequence length for RNAs between 500 nt and 1,000 nt.** The F-score is shown for pseudoknot prediction depending on the input sequence length. The performance of DivideFold, IPknot [16,17], ProbKnot [15], KnotFold [33], pKiss [8,9] and UFold [27] is reported here for RNAs between 500 and 1,000 nt on the Test dataset.

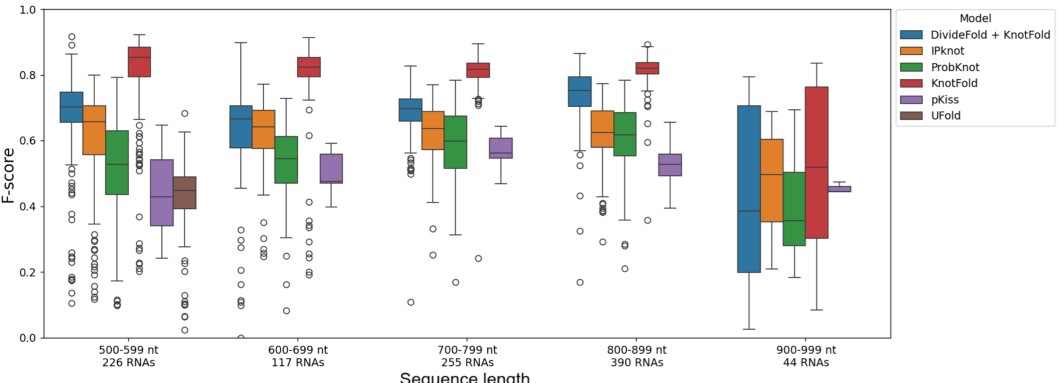

**Fig 16. Secondary structure prediction performance including pseudoknots by sequence length for RNAs between 500 nt and 1,000 nt.** The F-score is shown depending on the input sequence length. The performance of DivideFold, IPknot [16,17], ProbKnot [15], KnotFold [33], pKiss [8,9] and UFold [27] is reported here for RNAs between 500 and 1,000 nt on the Test dataset.

UFold only accepts sequences shorter than 600 nt. Considering the pseudoknot prediction, DivideFold still yields a better F-score than other considered tools for RNAs between 500 nt and 1,000 nt. Regarding the structure prediction including pseudoknots, despite showing satisfactory results, DivideFold is unable to perform better than KnotFold for RNAs between 500 nt and 1,000 nt. This makes DivideFold better suited for RNAs longer than 1,000 nt, which become increasingly difficult to deal with for other tools.

## Generalization to unseen families

We evaluate here the generalization to unseen families. Certain methods that are trained on data include benchmarks that sometimes consider a test dataset consisting only of different RNA families from the ones found in the training dataset [28,32,33], so that it can be seen if the tools manage to generalize to unseen families. Released in 2021, the bpRNA-new [32] dataset contains RNA families introduced in Rfam 14.2 [50] which were not yet discovered in Rfam 12.2 [51] when the bpRNA-1m [44] dataset was introduced. Thus, bpRNA-new only contains different families from bpRNA-1m, and is widely used in the literature as a test dataset to check for generalization capabilities to unseen families. However, sequences longer than 500 nt were removed from bpRNA-new, making it unsuitable for our approach. To solve this, we design a novel family-wise test dataset, which we name bpRNA-NF-15.0. For this, we followed the procedure that was used to create bpRNA-new. We gathered RNAs from newly identified families in the latest available version of Rfam at the time of writing, Rfam 15.0 [52], that were not yet introduced in Rfam 12.2 when bpRNA-1m was created, ensuring that bpRNA-NF-15.0 only contains different families from the ones in the Train dataset. Then, we checked for sequence similarity with CD-HIT-EST [48] at a 80% similarity threshold. In comparison to bpRNA-new, bpRNA-NF-15.0 is based on Rfam 15.0 rather than Rfam 14.2 and contains twice as many new unseen families. Furthermore, it contains sequences up to 951 nt long. Unfortunately, since the release of Rfam 12.2, no new families including sequences longer than 1,000 nt were discovered. This is a significant issue preventing us from properly studying the generalization capabilities of DivideFold to newly discovered RNA families, since DivideFold is better suited for RNAs longer than 1,000 nt. Nonetheless, we conduct here this analysis for RNA sequences between 500 and 1,000 nt, even though few sequences

**Table 5. Mean F-score comparison for pseudoknot prediction between the Test and bpRNA-NF-15.0 datasets.**

| Model | F-score on Test | F-score on bpRNA-NF-15.0 | F-score gap |
|---|---|---|---|
| DivideFold + KnotFold | **0.203** | **0.178** | -0.025 |
| KnotFold | 0.094 | 0.128 | 0.034 |
| IPknot | 0.003 | 0.042 | 0.039 |
| ProbKnot | 0.002 | 0.009 | 0.007 |
| pKiss | 0.000 | 0.088 | 0.088 |

The mean F-score for pseudoknot prediction of DivideFold, IPknot [16,17], ProbKnot [15], KnotFold [33] and pKiss [8,9] is reported here on the Test and bpRNA-NF-15.0 datasets, for RNAs between 500 and 1,000 nt.

**Table 6. Mean F-score comparison for secondary structure prediction including pseudoknots between the Test and bpRNA-NF-15.0 datasets.**

| Model | F-score on Test | F-score on bpRNA-NF-15.0 | F-score gap |
|---|---|---|---|
| DivideFold + KnotFold | 0.701 | 0.424 | -0.277 |
| KnotFold | **0.797** | 0.461 | -0.336 |
| IPknot | 0.616 | **0.464** | -0.152 |
| ProbKnot | 0.567 | 0.398 | -0.169 |
| pKiss | 0.493 | 0.395 | -0.098 |

The mean F-score for secondary structure prediction including pseudoknots of DivideFold, IPknot [16,17], ProbKnot [15], KnotFold [33] and pKiss [8,9] is reported here on the Test and bpRNA-NF-15.0 datasets, for RNAs between 500 and 1,000 nt.

in bpRNA-NF-15.0 are longer than 500 nt. For this analysis, we evaluate DivideFold, IPknot, ProbKnot, KnotFold and pKiss on bpRNA-NF-15.0 and we compare the results to those obtained on the Test dataset for the same sequence lengths. A maximum fragment length of 500 nt was used for DivideFold in this experiment. Implementation details for DivideFold on bpRNA-NF-15.0 are provided in the S1 File, along with more detailed results.

The performances on the Test and bpRNA-NF-15.0 datasets for RNAs between 500 and 1,000 nt are aggregated for comparison in Table 5 and Table 6 for pseudoknot prediction and secondary structure prediction including pseudoknots respectively.

Regarding the pseudoknot prediction, the performances on the Test and bpRNA-NF-15.0 datasets are similar, and DivideFold still yields the highest mean F-score in comparison to the other benchmarked tools.

Considering the secondary structure prediction including pseudoknots, the performance loss on bpRNA-NF-15.0 is highest for DivideFold and KnotFold, even though this is less significant for DivideFold, showing that the generalization to unseen families remains a serious concern for deep learning-based approaches. Notice however that the mean performance also decreases for IPknot, ProbKnot and pKiss, which are not trained methods, showcasing the difficulty of handling newly discovered families in general.

## Conclusion

DivideFold, which combines a recursive partition and deep learning techniques, offers a promising direction for enhancing the prediction of secondary structures including pseudoknots in long RNAs. Our approach addresses the challenges associated with lengthy RNA sequences by partitioning long sequences into shorter fragments through strategic cut point selection and leveraging existing structure prediction models designed for short RNAs. Our

approach maintains a linear time complexity in $O(n)$ and avoids the risk of memory overflows, making it capable of processing long RNAs.

We perform a benchmark in which we demonstrate the superior performance of our method compared to IPknot, ProbKnot and KnotFold for RNAs longer than 1,000 nt. DivideFold is able to achieve a significantly higher recall and precision for pseudoknot prediction in our experiments compared to the benchmarked methods. Regarding secondary structure prediction including pseudoknots, DivideFold exhibits a stronger performance overall as well. Nonetheless, our model struggles to find accurate cut points for very long RNAs beyond 2,500 nt.

We also attempt to check for generalization to new RNA families unseen in the Train dataset, but the absence of data for sequences longer than 1,000 nt belonging to newly discovered families prevents us from conducting a proper analysis for DivideFold. Nevertheless, we study this for RNAs between 500 and 1,000 nt and show that the performance drops significantly for all benchmarked tools, suggesting that predicting the structures and pseudoknots of RNAs from new families remains a tough challenge.

In the future, acquiring an increased number of very long RNA sequences and secondary structures, will be essential to enable more accurate predictions and comprehensively evaluate the model's performance. Finally, DivideFold will be able to be paired with any new accurate RNA secondary structure prediction method, benefiting from future research.

DivideFold, along with all the datasets used for this study, is accessible at https://evryrna.ibisc. univ-evry.fr/evryrna/dividefold/home.

## Supporting information

**S1 File A more in-depth analysis of the performance on bpRNA-NF-15.0 is provided.** The implementation details of DivideFold are given, along with detailed results for pseudoknot prediction and secondary structure prediction including pseudoknots of DivideFold, IPknot [16,17], ProbKnot [15], KnotFold [33], pKiss [8,9] and UFold [27]. (PDF)

## Author contributions

**Conceptualization:** Loïc Omnes, Fariza Tahi.

**Data curation:** Loïc Omnes.

**Funding acquisition:** Pierre Bartet, Fariza Tahi.

**Investigation:** Loïc Omnes.

**Methodology:** Loïc Omnes, Pierre Bartet, Fariza Tahi.

**Project administration:** Fariza Tahi.

**Supervision:** Eric Angel, Pierre Bartet, François Radvanyi, Fariza Tahi.

**Visualization:** Loïc Omnes.

**Writing – original draft:** Loïc Omnes.

**Writing – review & editing:** Loïc Omnes, Eric Angel, Pierre Bartet, François Radvanyi, Fariza Tahi.

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
