## [Decision Letter · Decision Letter 0]

9 Dec 2024

PONE-D-24-52867A divide-and-conquer approach based on deep learning for long RNA secondary structure prediction: focus on pseudoknots identificationPLOS ONE

Dear Dr. Tahi,

Thank you for submitting your manuscript to PLOS ONE. After careful consideration, we feel that it has merit but does not fully meet PLOS ONE’s publication criteria as it currently stands. Therefore, we invite you to submit a revised version of the manuscript that addresses the points raised during the review process.

We look forward to receiving your revised manuscript.

Kind regards,

Emanuele Paci

Academic Editor

PLOS ONE

Journal Requirements:

2. Please expand the acronym “GENCI/IDRIS” (as indicated in your financial disclosure) so that it states the name of your funders in full. This information should be included in your cover letter; we will change the online submission form on your behalf.

3. Thank you for stating the following in the Acknowledgments Section of your manuscript: [This work was supported by grants from Région Ile-de-France. This work was granted access to the HPC resources from GENCI/IDRIS (grant AD011014250)] We note that you have provided funding information that is not currently declared in your Funding Statement. However, funding information should not appear in the Acknowledgments section or other areas of your manuscript. We will only publish funding information present in the Funding Statement section of the online submission form. Please remove any funding-related text from the manuscript and let us know how you would like to update your Funding Statement. Currently, your Funding Statement reads as follows: [LO was supported by grants from Région Ile-de-France (Paris Region PhD 2022, https://www.iledefrance.fr/aides-et-appels-a-projets/paris-region-phd-2024) and GENCI/IDRIS (grant AD011014250, http://www.idris.fr/).

The funders did not play any role in the study design, data collection and analysis, decision to publish, or preparation of the manuscript.] Please include your amended statements within your cover letter; we will change the online submission form on your behalf.

Additional Editor Comments:

Dear Dr Tahi,

Please find the reviews below. While your manuscript is publishable on PLoS ONE, I ask you to address the enclosed reviews, particularly that of reviewer 1 asking for major revision.

Kind regards

Emanuele Paci

Reviewers' comments:

Reviewer's Responses to Questions

**Comments to the Author**

1. Is the manuscript technically sound, and do the data support the conclusions?

Reviewer #1: Partly

Reviewer #2: Yes

2. Has the statistical analysis been performed appropriately and rigorously? 

Reviewer #1: Yes

Reviewer #2: Yes

3. Have the authors made all data underlying the findings in their manuscript fully available?

Reviewer #1: Yes

Reviewer #2: Yes

4. Is the manuscript presented in an intelligible fashion and written in standard English?

Reviewer #1: Yes

Reviewer #2: Yes

5. Review Comments to the Author

Reviewer #1: This paper proposes a novel divide-and-conquer method, DivideFold, to address the challenges of predicting RNA secondary structures, particularly pseudoknots, in long sequences. DivideFold recursively partitions RNA sequences into smaller fragments, applies existing structure prediction models to each fragment, and reassembles the overall structure. The method demonstrates superior performance in pseudoknot prediction compared to existing models, with significant computational efficiency for long RNA sequences. Furthermore, the authors provide open access to the DivideFold source code and datasets, enhancing the reproducibility of the research.

The paper splits the dataset based on sequence similarity, but this approach poses a risk of overfitting. RNA sequences within the same family often share characteristic structural patterns, and splitting solely based on sequence similarity can lead to an overlap between the training and test sets, artificially inflating the model's performance.

To address this issue, it is recommended to split the data based on RNA families. For instance, if bpRNA_1m is used for training, an independent dataset such as bpRNAnew dataset (Sato et al, 2021) should be used for testing. This approach would provide a more realistic evaluation of the model's generalization capabilities, particularly for unseen RNA families.

In DivideFold, detecting cut points for splitting sequences into inner and outer fragments requires predicting two cut points simultaneously. This necessitates evaluating the relationships between all pairs of nucleotides, leading to a computational complexity of at least O(n^2).

The paper claims that the cut point detection algorithm operates with O(n) complexity, but this seems to omit the simultaneous prediction of inner and outer cut points or rely on simplifications. A more detailed explanation of the algorithm and its computational complexity is necessary for clarity and validation.

The paper describes the removal of pseudoknots before cut point selection, but it does not clarify how the nucleotide pairs involved in pseudoknots ((i, j) and (k, l) for i<k<j<l) are="" common="" criteria="" for="" include:="" might="" pseudoknot="" removal="" removal.="" selected="selected">- Minimizing free energy for thermodynamic stability,

- Prioritizing longer or more stable stems,

- Heuristic approaches to preserve the overall structure.

The criteria for removal should be clearly explained, because which base pairs to remove would affect the accuracy of the prediction.</k<j<l)>

Reviewer #2: The accurate prediction of RNA secondary structures is still a great challenge, especially for RNAs with long sequence and/or pseudoknots. This work proposed a novel approach to solve this problem by partitioning a long sequence into shorter ones. For this approach, the key is how to maintain the stems formed by long-range bases. The most novel point of this work is proposing a method to ensure that the long-range pairing is not disrupted by the partition. The benchmarks show that in most cases their method overperforms the existing methods in prediction of secondary structure and pseudoknots for long RNAs, including computation time. The results may also be important to understand the mechanism of RNA secondary structures. This paper is clearly written too. I recommend it for publication after minor revisions.

1. Line 151: “For each motif, we check for matches in the sequence through regular expressions and count the number of possible configurations for the motif to be inserted at each point in the sequence.” Give details using a picture

2. It is better to give the average performance of cut-point prediction because this is the key to this approach.

3. It is interesting to know how about the performance of this approach for middle length, e.g., 500-1000nt. Suggest giving a brief discussion.

6. PLOS authors have the option to publish the peer review history of their article (what does this mean?). If published, this will include your full peer review and any attached files.

Reviewer #1: No

Reviewer #2: No

---

## [Author Response · Author response to Decision Letter 1]

19 Dec 2024

Dear editor,

We believe that this revised version answers to the issues that were raised by you and the reviewers.

Our answers to the reviewers' comments can be found in the Response to Reviewers file.

We thank you and the reviewers for their valuable comments.

Regarding the updated funding statement, it should be as is now written in the cover letter :

LO was supported by grants from Région Ile-de-France (Paris Region PhD 2022, https://www.iledefrance.fr/aides-et-appels-a-projets/paris-region-phd-2024) and Grand Equipement National de Calcul Intensif (GENCI) / Institut du Développement et des Ressources en Informatique Scientifique (IDRIS) (grant AD011014250, http://www.idris.fr/). The funders did not play any role in the study design, data collection and analysis, decision to publish, or preparation of the manuscript.

Kind regards,

Fariza Tahi

---

## [Decision Letter · Decision Letter 1]

28 Jan 2025

PONE-D-24-52867R1A divide-and-conquer approach based on deep learning for long RNA secondary structure prediction: focus on pseudoknots identificationPLOS ONE

Dear Dr. Tahi,

Thank you for submitting your manuscript to PLOS ONE. After careful consideration, we feel that it has merit but does not fully meet PLOS ONE’s publication criteria as it currently stands. Therefore, we invite you to submit a revised version of the manuscript that addresses the points raised during the review process.

We look forward to receiving your revised manuscript.

Kind regards,

Emanuele Paci

Academic Editor

PLOS ONE

Journal Requirements:

Additional Editor Comments:

One of the reviewers has still major concerns on the cross validation of your results. If you cannot fulfill their request, which seems to me justified, please make it clear that your results may be seriously affected by the procedure you used and should be taken with caution.

Reviewers' comments:

Reviewer's Responses to Questions

**Comments to the Author**

1. If the authors have adequately addressed your comments raised in a previous round of review and you feel that this manuscript is now acceptable for publication, you may indicate that here to bypass the “Comments to the Author” section, enter your conflict of interest statement in the “Confidential to Editor” section, and submit your "Accept" recommendation.

Reviewer #1: All comments have been addressed

Reviewer #2: All comments have been addressed

2. Is the manuscript technically sound, and do the data support the conclusions?

Reviewer #1: Partly

Reviewer #2: (No Response)

3. Has the statistical analysis been performed appropriately and rigorously? 

Reviewer #1: Yes

Reviewer #2: (No Response)

4. Have the authors made all data underlying the findings in their manuscript fully available?

Reviewer #1: Yes

Reviewer #2: (No Response)

5. Is the manuscript presented in an intelligible fashion and written in standard English?

Reviewer #1: Yes

Reviewer #2: (No Response)

6. Review Comments to the Author

Reviewer #1: (Sato et al., 2021) and many other papers have pointed out that it is not possible to detect overfitting using only sequence-wise cross-validation with tools such as CD-HIT. In fact, some of the papers you cited as examples of sequence-wise cross-validation in your rebuttal believe that sequence-wise cross-validation alone is not sufficient, and they also conduct evaluation using family-wise cross-validation. To give an example, E2Efold (Chen et al., ICLR 2020) was evaluated for accuracy using only sequence-wise cross-validation, and while it achieved good accuracy there, subsequent validation by several papers by other groups pointed out that it fell into severe overfitting (e.g. Sato et al., 2021; Foo et al., 2022). Therefore, the current situation, where accuracy is evaluated only by sequence-wise cross-validation, cannot be acceptable. If the evaluation using the bpRNAnew dataset that I requested is inappropriate due to the problem of sequence length, you should construct an appropriate dataset to conduct the family-wise cross-validation.

Reviewer #2: (No Response)

7. PLOS authors have the option to publish the peer review history of their article (what does this mean?). If published, this will include your full peer review and any attached files.

Reviewer #1: No

Reviewer #2: No

---

## [Author Response · Author response to Decision Letter 2]

24 Feb 2025

Dear reviewers,

We thank you for your valuable comment.

We have corrected our manuscript and provided a supplementary file to answer it.

We have built a new dataset, bpRNA-NF-15.0, to check for generalization to unseen RNA families for longer RNAs.

Even though bpRNA-NF-15.0 does not contain any RNAs longer than 1000 nt, making it difficult to properly evaluate our model, we conducted an analysis on sequences between 500 and 1000 nt, which were absent from the existing dataset bpRNA-new.

Sincerely,

---

## [Editor Report · Decision Letter 2]

4 Mar 2025

A divide-and-conquer approach based on deep learning for long RNA secondary structure prediction: focus on pseudoknots identification

PONE-D-24-52867R2

Dear Dr. Tahi,

We’re pleased to inform you that your manuscript has been judged scientifically suitable for publication and will be formally accepted for publication once it meets all outstanding technical requirements.

Kind regards,

Emanuele Paci

Academic Editor

PLOS ONE
---

## [Editor Report · Acceptance letter]

PONE-D-24-52867R2

PLOS ONE

Dear Dr. Tahi,

I'm pleased to inform you that your manuscript has been deemed suitable for publication in PLOS ONE. Congratulations! Your manuscript is now being handed over to our production team.

Kind regards,

on behalf of

Dr. Emanuele Paci

Academic Editor

PLOS ONE